# Dynamically prognosticating patients with hepatocellular carcinoma through survival paths mapping based on time-series data

Lujun Shen[1,2], Qi Zeng[3], Pi Guo[4], Jingjun Huang[5], Chaofeng Li[2,6], Tao Pan[7], Boyang Chang[1,2], Nan Wu[8], Lewei Yang[3], Qifeng Chen[1,2], Tao Huang[1,2], Wang Li[1,2] & Peihong Wu[1,2]

Patients with hepatocellular carcinoma (HCC) always require routine surveillance and repeated treatment, which leads to accumulation of huge amount of clinical data. A predictive model utilizes the time-series data to facilitate dynamic prognosis prediction and treatment planning is warranted. Here we introduced an analytical approach, which converts the time-series data into a cascading survival map, in which each survival path bifurcates at fixed time interval depending on selected prognostic features by the Cox-based feature selection. We apply this approach in an intermediate-scale database of patients with BCLC stage B HCC and get a survival map consisting of 13 different survival paths, which is demonstrated to have superior or equal value than conventional staging systems in dynamic prognosis prediction from 3 to 12 months after initial diagnosis in derivation, internal testing, and multicentric testing cohorts. This methodology/model could facilitate dynamic prognosis prediction and treatment planning for patients with HCC in the future.

[1] Department of Minimally Invasive Interventional Therapy, Sun Yat-sen University Cancer Center, Guangzhou 510060 Guangdong, China. [2] State Key Laboratory of Oncology in South China, Collaborative Innovation Center for Cancer Medicine, Guangzhou 510060 Guangdong, China. [3] Department of Radiation Oncology, Fifth Affiliated Hospital of Sun Yat-sen University, Zhuhai 519000 Guangdong, China. [4] Department of Preventive Medicine, Shantou University Medical College, Shantou 515063 Guangdong, China. [5] Department of Minimally Invasive Interventional Radiology, Second Affiliated Hospital of Guangzhou Medical University, Guangzhou 510260 Guangdong, China. [6] Information Center, Sun Yat-sen University Cancer Center, Guangzhou 510060 Guangdong, China. [7] Department of Vascular Interventional Radiology, Third Affiliated Hospital of Sun Yat-sen University, Guangzhou 510530 Guangdong, China. [8] Department of Family Medicine, Memorial University of Newfoundland, St. John's A1C5S7 Newfoundland and Labrador, Canada. These authors contributed equally: Lujun Shen, Qi Zeng, Pi Guo, Jingjun Huang.  Correspondence and requests for materials should be addressed to W.L. (email: liwang@sysucc.org.cn) or to P.W. (email: wuph@sysucc.org.cn)

Hepatocellular carcinoma (HCC) is the sixth most common cancer and the third leading cause of cancer-related death worldwide[1], characterized by multicentric origins, high risk of intrahepatic recurrence, and poor prognosis[2]. For most patients with intermediate-stage HCC, comprehensive treatments using transarterial chemoembolization (TACE), ablative therapies, surgery, or their combinations were widely adopted; besides, frequent disease surveillance and re-treatment are needed[3–6]. The current dynamic prognostication systems, including hepatoma arterial-embolization prognostic score and assessment for re-treatment with TACE (ART) score were designed specifically for patients receiving arterial-embolization therapies[7,8]. A novel prognostication system that suitable for HCC patients receiving comprehensive treatments and facilitate dynamic treatment planning is desirable.

During the process of ongoing surveillance and treatment, the time-series clinical data rapidly accumulate. The data generated from HCC patients are multimodal, frequently obtained at regular interval, and whose impact may not be same in patients' life course[9,10]. Understanding these data may delineate the biological behaviors of HCC and help guide dynamic management[11].

In the past decade, extensive studies had been conducted to understand the value of the time-series clinical data. Several studies have been reported that the joint models can describe the change in the prognostic value of a single variable measured over time[12,13], while these models do not support multiple dynamic variables, which limit its clinical application. In the field of machine learning, the methodologies of temporal abstractions[14] and hidden Markov models[15] had been utilized for classifying patients with longitudinal data by building splitting trajectories; these models supported multiple variables and achieved better performance compared to modeling just using cross-sectional data. However, these models are considered opaque since internal structure and learned parameters are difficult for interpretation. Moreover, pure pursuit of the precision in prognosis prediction it not that important as an ideal predictive system should also provide clues in treatment planning[2].

Patients with HCC are regularly followed up every 1–3 months during the treatment, every 3 months during the first year after complete remission, and later every 3–6 months[2]. Therefore, in this study, we converted the time-series data of patients with Barcelona clinic liver cancer (BCLC) stage B HCC into data of time slices with constant interval of 3 months, and used a process of "Cox-based feature selection" to select key prognostic features to build the first cascading survival map. The survival paths established present superior or equal ability in dynamic prognosis prediction than the conventional staging systems from the time frame of 3–12 months after diagnosis. Based on the map, we identify three paths of long-term survival; three chance nodes with progressive disease but high chance (>80%) to have improved survival after surgery/ablation; one incurable node with progressive disease, poor median overall survival (OS), and no transferred path. The methodology of survival path mapping can be utilized to facilitate dynamic prognosis prediction and treatment planning for patients with HCC in the future.

## Results

**The baseline characteristic of the patients.** The derivation cohort consisted of 979 HCC patients with a median age of 55 (range 15–86) years, the internal testing cohort consisted of 627 HCC patients with a median age of 56 (range 20–89), and the multicenter testing cohort consisted of 414 patients with a median age of 52 (range 19–80). The multicenter testing cohort had a higher proportion of female and young patients than the derivation cohort; there were no significant differences in hepatitis B virus infection, serum alpha-fetoprotein (AFP) level, Child-Pugh class, tumor size, or number of lesions between the two testing cohorts and derivation cohort (Table 1).

**The survival paths built by derivation cohort.** Using 3 months as the interval of each time slice, the first 2 years' dataset of derivation cohort was converted into the data of nine time slices, which includes 979, 822, 513, 390, 336, 294, 246, 221, and 202

**Table 1 Baseline characteristics of derivation, validation, and testing cohort at initial diagnosis**

| Variable | Derivation cohort no. (%) | Internal testing cohort no. (%) | P value | Multicenter testing cohort no. (%) | P value |
|---|---|---|---|---|---|
| Age (years) | | | 0.441 | | <0.001 |
| ≤50 | 370 (37.8) | 249 (39.7) | | 199 (48.1) | |
| >50 | 609 (62.2) | 378 (60.3) | | 215 (51.9) | |
| Gender | | | 0.113 | | 0.043 |
| Male | 889 (90.8) | 554 (88.4) | | 361 (87.2) | |
| Female | 90 (9.2) | 73 (11.6) | | 53 (12.8) | |
| HBV infection | | | 0.086 | | 0.813 |
| No | 33 (3.4) | 32 (5.1) | | 15 (3.6) | |
| Yes | 946 (96.6) | 595 (94.9) | | 399 (96.4) | |
| AFP (IU/ml) | | | 0.493 | | 0.314 |
| <25 | 318 (32.5) | 214 (34.1) | | 146 (35.3) | |
| ≥25 | 661 (67.5) | 413 (65.9) | | 268 (64.7) | |
| Child-Pugh class | | | 0.815 | | 0.602 |
| A | 841 (85.9) | 536 (85.5) | | 360 (87.0) | |
| B | 138 (14.1) | 91 (14.5) | | 54 (13.0) | |
| Tumor size (cm) | | | 0.727 | | 0.939 |
| Mean ± SD | 7.20 ± 3.57 | 7.07 ± 3.48 | | 7.12 ± 3.51 | |
| ≤5 | 329 (33.6) | 216 (34.4) | | 140 (33.8) | |
| >5 | 650 (66.4) | 411 (65.6) | | 274 (66.2) | |
| Number of lesions | | | 0.428 | | 0.956 |
| ≤3 | 391 (39.9) | 238 (38.0) | | 166 (40.1) | |
| >3 | 588 (60.1) | 389 (62.0) | | 248 (59.9) | |

All values are presented as numbers of patients followed by percentages in the parentheses. P values were calculated by comparing categorical variables between testing cohorts and derivation cohort with chi-square test

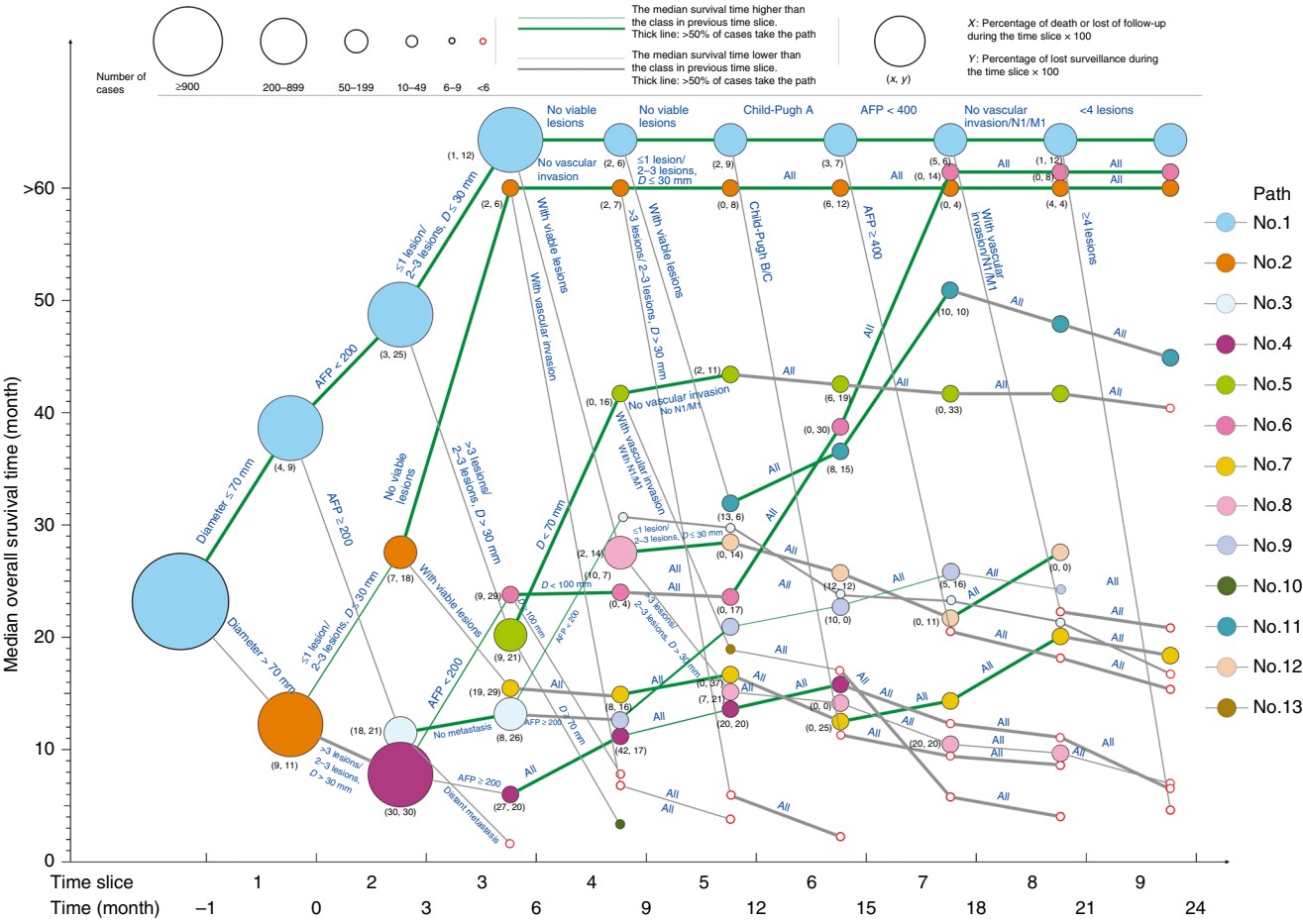

**Fig. 1** The survival path system constructed for BCLC stage B HCC patients. Using the selected features identified at each time slice, the population was divided into cascades of subgroups, which was further visualized by two-dimensional graph, with the time slices on x-axis and median OS time on y-axis. A total of 13 different paths were constructed

cases with effective data, respectively; the significance value α was set at 0.006. After completing all the processing cycle of the derivation cohort, the data from time slice 1–9 were divided into 2, 4, 7, 10, 12, 12, 10, 10, and 7 subclasses, respectively; subclasses with <6 cases were excluded from the mapping to reduce the risk of model overfitting. By connecting the class with its derivative classes, a total of 13 survival paths were constructed, which were illustrated in different colors (Fig. 1).

In this model of survival path, every bifurcation point is called a node, and each node integrates the information of previous nodes to facilitate prognosis prediction.

**Prognostic value of survival path in derivation cohort**. In the derivation cohort, the prognostic power between the survival path system, BCLC staging system, AJCC staging system, and ART score system was compared at all nine time slices (Table 2). The survival path system had superior or non-inferior c-index in predicting OS than BCLC staging system, AJCC staging system, and ART score system from time slice No.3 to time slice No.9. At time slice No.2, the AJCC staging system had superior c-index than survival path system, while no significant difference in c-index between the survival path system and BCLC staging system was found. It was interesting to note that the survival differences between stage B and stage C in BCLC staging system, as well as the differences between stage IIIa, stage IIIb, and stage IVb in AJCC staging system, diminished to insignificance starting at

time slice No.5; by contrast, the survival path system presented superior performance in dynamically discrimination the OS of HCC patients (Fig. 2).

**Validation of the survival path system in testing cohorts**. Generally, the survival curves in the internal testing cohort and multicenter testing cohort fit well with the curves in the derivation cohort (Fig. 3). In the internal testing cohort, the survival path system showed superior and non-inferior prognostic value than the BCLC staging system and AJCC staging system from time slice No.3 to time slice No.5. The advantages of the survival path system diminished starting at time slice No.6. The results in the multicenter testing cohort confirmed the advantages of the survival path system over other two staging systems from time slice No.3 to time slice No.5 (Table 3). The significance of each path bifurcation was also evaluated in the testing cohorts (Table 4). A P value < 0.05 could be achieved in all the bifurcation with enough (≥6) cases in each of the following comparator nodes, which demonstrated the stability of the survival path system we built.

**Long-term survival based on the survival paths system**. Of the 13 paths constructed, 3 paths lead to long-term survival (>60 months), including No.1, No.2, and No.8. The No.1 and No.2 paths reached long-term survival when no viable lesions was achieved. However, disease progression could occur at any time

**Table 2 Comparison of *c*-index between the survival path system, BCLC staging system, AJCC staging system, and ART score at each time slice in the derivation cohort**

| Time slice | Number of cases (modeling/all)[a] | Survival path system | | BCLC staging system | | AJCC staging system | | ART class | |
|---|---|---|---|---|---|---|---|---|---|
| | | Number of nodes | *c*-Index (95% CI) | Number of classes | *c*-Index (95% CI) | Number of classes | *c*-Index (95% CI) | Number of classes | *c*-Index (95% CI) |
| No.1 | 979/979 | 2 | 0.624 (0.623–0.625) | 1 | — | 2 | 0.602 (0.601–0.603) | 1 | — |
| No.2 | 822/822 | 4 | 0.695 (0.693–0.697) | 5 | 0.696 (0.694–0.698) | 6 | 0.702[c] (0.702–0.704) | 2 | 0.528[b] (0.526–0.530) |
| No.3 | 506/513 | 7 | 0.733 (0.730–0.736) | 4 | 0.725[b] (0.722–0.728) | 6 | 0.733 (0.730–0.736) | 2 | 0.536[b] (0.533–0.539) |
| No.4 | 374/390 | 10 | 0.760 (0.756–0.764) | 4 | 0.724[b] (0.720–0.728) | 6 | 0.727[b] (0.723–0.731) | 2 | 0.572[b] (0.568–0.576) |
| No.5 | 307/336 | 12 | 0.768 (0.763–0.773) | 4 | 0.731[b] (0.726–0.736) | 6 | 0.737[b] (0.731–0.743) | 2 | 0.589[b] (0.584–0.594) |
| No.6 | 245/294 | 12 | 0.771 (0.764–0.778) | 5 | 0.749[b] (0.742–0.756) | 6 | 0.757[b] (0.750–0.764) | 2 | 0.535[b] (0.528–0.542) |
| No.7 | 199/246 | 10 | 0.792 (0.783–0.801) | 4 | 0.764[b] (0.755–0.773) | 6 | 0.766[b] (0.756–0.776) | 2 | 0.563[b] (0.554–0.572) |
| No.8 | 167/221 | 10 | 0.811 (0.799–0.823) | 4 | 0.773[b] (0.762–0.784) | 6 | 0.769[b] (0.757–781) | 2 | 0.541[b] (0.530–0.550) |
| No.9 | 128/202 | 7 | 0.830 (0.816–0.844) | 4 | 0.769[b] (0.752–0.786) | 6 | 0.802 (0.785–0.819) | 2 | 0.549[b] (0.531–0.567) |

[a]Nodes of survival path system with less than six cases were excluded from the computing of *c*-index. Therefore, the number of cases in modeling is less than the number of all cases with effective data
[b]The *c*-index of the interested system was lower than survival path system, with $P < 0.006$
[c]The *c*-index of the interested system was higher than the survival path system, with $P < 0.006$

slice in a small proportion of patients even they are on the paths of long-term survival. For the No.8 path, due to the limitation of our sample size, the key factors related to long-term survival fail to be identified.

**Treatment and the path transfer**. Of all the nodes in the survival path system, five nodes went down from bifurcated nodes in the previous time slice and bifurcated in the following time slice. These nodes had unfavorable prognosis and the survival path system might provide guidance. Surgery and ablative therapies are considered aggressive management and therefore we described the proportion of patients receiving surgery/ablation in these nodes (Table 5). The surgery/ablation rates in $S_{(p=3, \text{ts}=2)}$, $S_{(p=5, \text{ts}=3)}$, $S_{(p=8, \text{ts}=4)}$, $S_{(p=2, \text{ts}=1)}$, and $S_{(p=4, \text{ts}=2)}$ were 23.3%, 24.5%, 31.4%, 25.2%, and 13.2%, respectively; candidates who received surgery/ablation had rates of 83.3%, 84.6%, 81.3%, 71.2% and 56.7% going to the upper node in the next time slice, respectively. We define a node meets both following conditions: (1) median OS time of its upper bifurcated node had 10 months higher than that of the lower bifurcated node; (2) more than 80% patients receiving surgery/ablation went to the upper bifurcated node, as a chance node; then the $S_{(p=3, \text{ts}=2)}$, $S_{(p=5, \text{ts}=3)}$, and $S_{(p=8, \text{ts}=4)}$ are candidate nodes.

**Incurable disease based on the survival paths system**. Of all the paths in this map, two paths (No.10 and No.13) only have one valid node after bifurcation. We define a node without following valid node after bifurcation and had a median OS time less than 5 months as an incurable node, then $S_{(p=10, \text{ts}=4)}$ is the incurable node.

**Discussion**

In this work, we developed an analytical model to dynamically trace the prognosis of cancer patients with BCLC stage B HCC. Time slice was employed for data conversion and Cox-based feature selection for constructing the cascade structure of survival path. The survival path model showed superior or non-inferior prognostic value than the conventional BCLC and AJCC staging system from time slice No.3 to time slice No.5, which were confirmed in internal and multicenter testing cohorts. These results suggest that this tool is valuable in dynamic prognosis prediction and treatment planning for patients with intermediate-stage HCC during the time frame of 3–12 months after diagnosis.

Currently, studies on developing re-staging systems for malignant tumors were always hindered by the lack of effective methods in utilizing with time-series data, and by the fact that different treatments could lead to different re-staging strategies, which restrict the generalization of established models[16–18]. In HCC, the BCLC staging system is most widely used for staging and re-staging during clinical practice. This classification uses variables related to tumor stage, liver functional status, physical status, and cancer-related symptoms, and has been validated as the best staging system for treatment guidance[2]. However, the provided information of BCLC staging system was not enough to support dynamic prognosis prediction and real-time treatment planning, as reflected by our results that the survival differences between Stage B and Stage C subgroups gradually decreased over time. Hence we proposed to create a more precise system. The survival path we built for intermediate-stage HCC integrated the time-series information of variables utilized in BCLC classifications, variables on image change after treatment[19], and variables on important serum markers[20,21]; although only one selected feature was utilized for node subdivision at each time slice, the model constructed showed superior or non-inferior prognostic value than the BCLC staging system at all time slices in the derivation cohort, indicating that this methodology had great potential.

In the testing cohorts, we observed that the *c*-index of survival path system decreased at time slice No.6 and the advantages of the survival path system compared to BCLC and AJCC staging systems diminished starting at time slice No.6. Moreover, the *c*-index of BCLC staging system was significantly higher than the survival path system in the internal testing cohort at time slice No.6. These phenomena may be caused by the fact that no more path bifurcations were made since time slice No.6 due to the

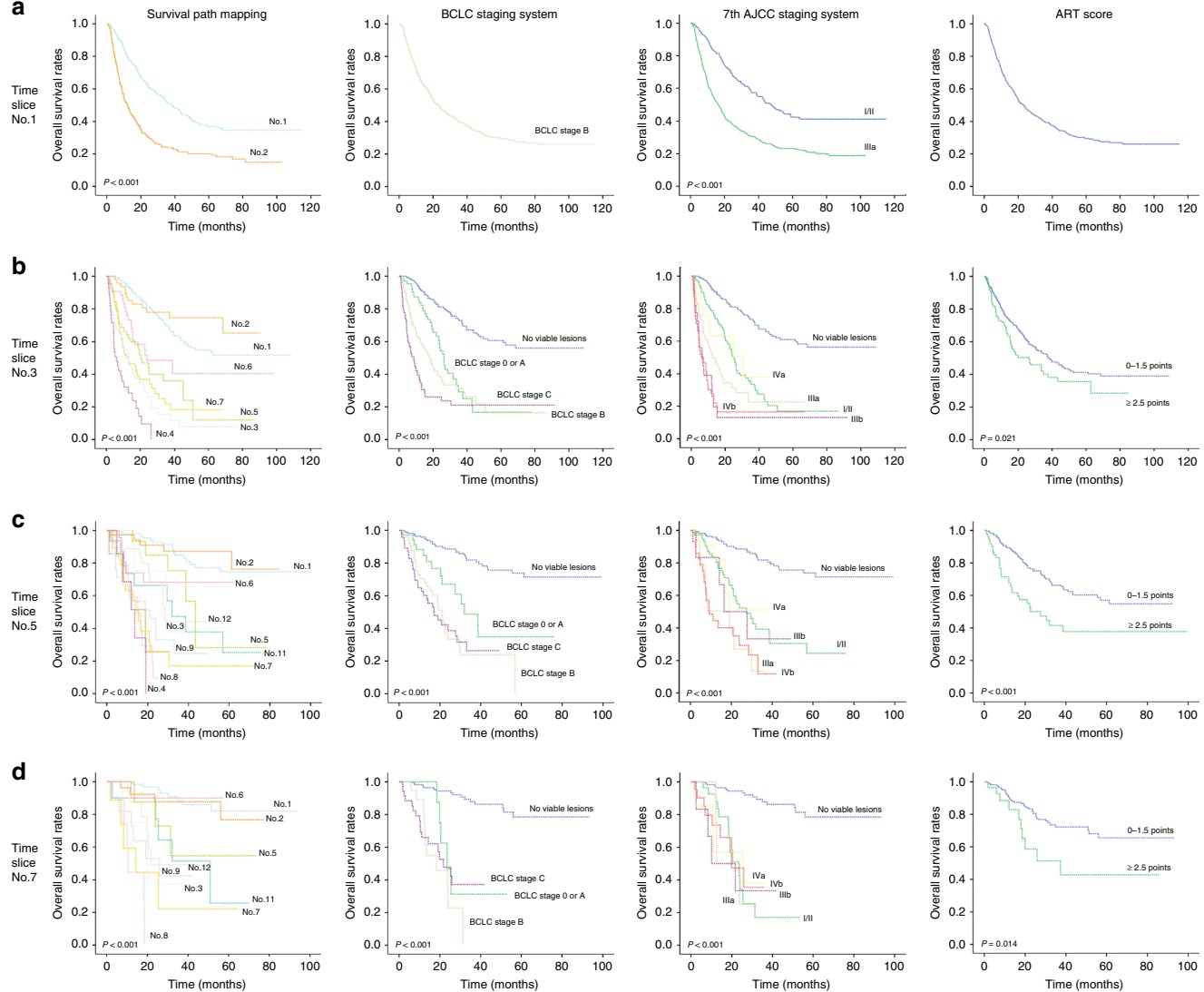

**Fig. 2** Kaplan–Meier plots in the derivation cohort. Kaplan–Meier plots showed OS divided by the survival path system, BCLC staging system, AJCC staging system, and ART score, respectively, at time slice No.1 (**a**), time slice No.3 (**b**), time slice No.5 (**c**), and time slice No.7 (**d**) in the derivation cohort

restriction of the sample size, which impaired the ability of survival path system in utilizing the latest information of patients. It is conceivable that the predictive value of this system could further be enhanced if we can utilize a big clinical database of thousands of cases, as more refined paths can be constructed and the data in distant time slices can be well analyzed.

The survival path model we established can also guide treatment planning for intermediate-stage HCC. In the management of patients on the paths going up, we need to closely pay attention to the key variables that could transfer the patient to unfavorable paths in the following time slices. For example, for patient in $S_{(p=1,\ ts=2)}$, it is recommended to control the disease with $\leq 1$ viable lesion or 2–3 lesions with maximal tumor size <30 mm by the end of time slice No.3. Complete remission, AFP < 400 ng/ml, and Child-Pugh score A are required to maintain the patient on the No.1 path. In dealing of patients with progressive disease, aggressive treatments like surgery or ablative therapies could be utilized for those in chance nodes, while palliative treatments were recommended for patients in incurable nodes.

The survival paths we constructed for stage B HCC was an initial attempt, and efforts could be made in several aspects to

further improve this model/methodology. The first aspect is to use learning algorithms to explore the best cutoff for individual variable and optimizing the process of feature selection, including random forest[22], $k$-nearest neighbor[23], and neural networks[24]; bootstrap validation could be utilized alongside to ensure quality control and reduce the risk of overfitting. The second aspect is to develop algorithms for node fusion, which may enhance our utilization of the database and give us an insight into the biological behavior of cancer. The third aspect is to develop methods in dealing with irregular time series. Converting the time-series data into time slices of 3 months will result in some missing data, therefore, a learning algorithm that dynamically design the interval of time slice in specific node based on the characteristics of data is needed to maximally utilize the data.

In conclusion, the survival path model constructed in this study offers a superior method for dynamic prognostication for HCC patients during the time frame of 3–12 months after diagnosis, compared with the current BCLC staging system. The methodology utilized in this study also pioneers as an effective tool in processing the clinical big data of cancer patients in the future.

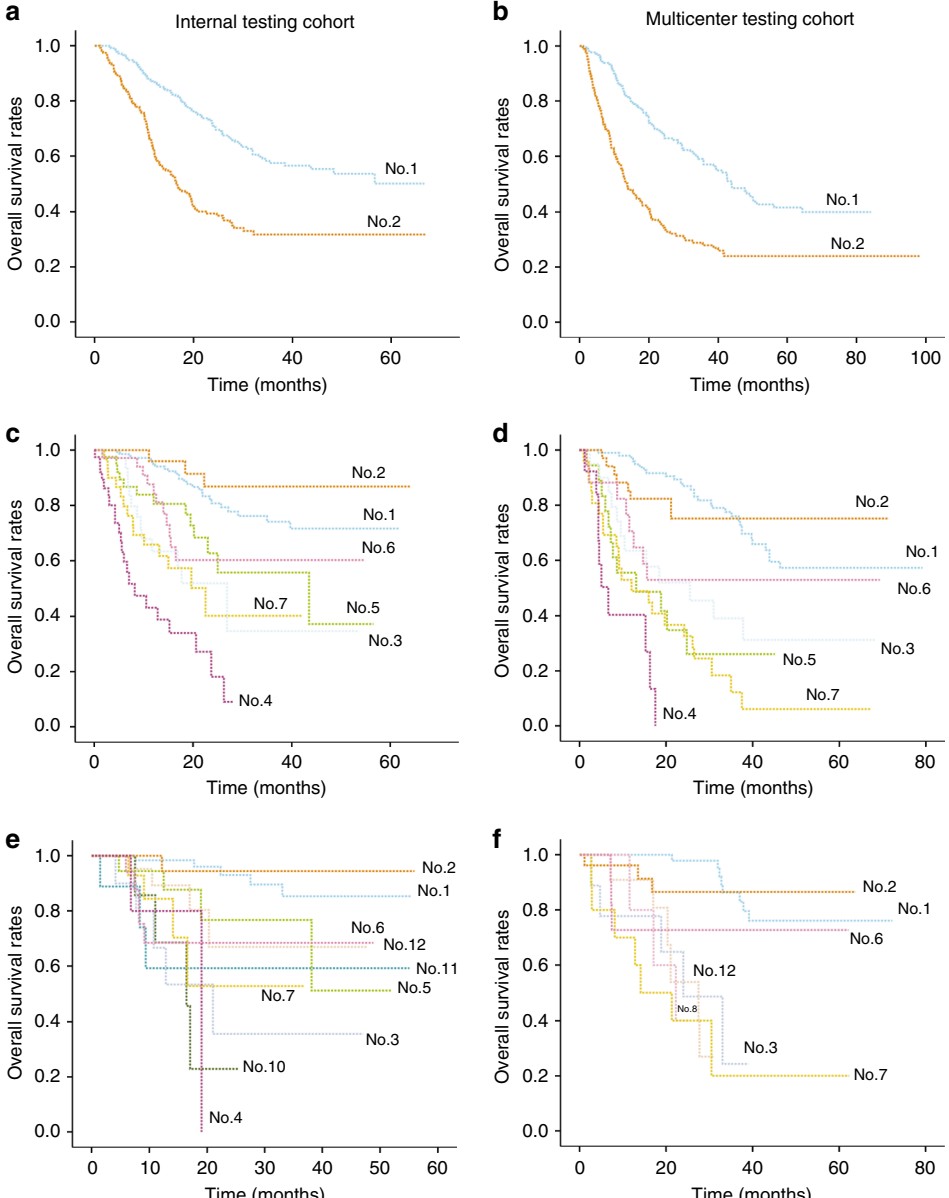

**Fig. 3** Kaplan–Meier plots in the validation cohorts. Kaplan–Meier plots showed OS divided by the survival path system in the internal testing cohort and multicenter testing cohort, respectively, at time slice No.1 (**a**, **b**), time slice No.3 (**c**, **d**), and time slice No.5 (**e**, **f**). Note: nodes of paths with <6 cases in the testing cohorts were regarded unstable and not included in the analysis

## Methods

**Patients and variables of interest**. Between January 2007 and May 2012, 5005 consecutive patients with newly diagnosed HCC at Sun Yat-sen University Cancer Center (SYSUCC) were retrospectively reviewed to develop the derivation cohort. Between June 2012 and December 2015, an independent consecutive series of 3843 HCC patients treated at SYSUCC were reviewed to develop the internal testing cohort. Besides, between January 2010 and December 2016, 843 patients from Fifth Affiliated Hospital of Sun Yat-sen University, 415 patients from the Third Affiliated Hospital of Sun Yat-sen University, and 437 patients from the Second Hospital of Guangzhou Medical University were reviewed to develop the multicenter testing cohort. The inclusion criteria were as follows: (1) clinically diagnosed with BCLC stage B HCC; (2) complete data of any of the following at initial diagnosis: computed tomography (CT) or magnetic resonance imaging (MRI) of the abdominal region, radiography or CT of the chest, routine bloodwork test, biochemical routine test, serum AFP level, and coagulation indices; and (3) without history of other malignancies. A total of 979, 627, and 414 patients were included in the derivation cohort, internal testing cohort, and multicenter testing cohort, respectively.

The Department of Clinical Research of Sun Yat-sen University Cancer Center approved the study protocol (2017-FXY-129). The Hospital Ethics Committee of

the four medical centers approved this study, which waived the need for written informed consent based on the retrospective nature of the study.

Most patients (1852/2020, 91.7%) in the above-mentioned medical centers received TACE as their first-line treatment; 135 (6.7%) patients received surgical resection as initial treatment and 33 (1.6%) patients refused to received treatment. The subsequent therapies after TACE, which constituted of ablative therapies, surgical resection, targeted therapies, or palliative chemotherapy, were adopted based on the decision of the multidisciplinary teams, including hepatologists, radiologists, and interventional radiologists. Patients were followed up monthly during the period of initial treatment, subsequently at every 2–3 months for the first 2 years if complete remission was achieved. The frequency gradually decreased to every 3–6 months after 2 years' remission.

Time-series data on serum tumor markers, biochemical and hematological indices, medical imaging, and associated changes (CT and/or MRI) of each patient were collected. Based on past literatures on staging systems and laboratory tests[2,25], a total of nine variables were designed, covering imaging results, laboratory tests, and performance status; all the variables were dichotomized (Table 6).

**Transformation of datasets for analysis**. To analyze the data, the time-series data were grouped into data at standard time slices. For every patient, the zero

**Table 3 Comparison of *c*-index between the survival path system, BCLC staging system, and AJCC staging system in the internal testing cohort and multicenter testing cohort**

| Time slice | Number (modeling/all) | Survival path system | | BCLC staging system | | AJCC staging system | |
|---|---|---|---|---|---|---|---|
| | | Number of nodes | *c*-Index (95% CI) | Number of classes | *c*-Index (95% CI) | Number of classes | *c*-Index (95% CI) |
| **Internal testing cohort** | | | | | | | |
| No.1 | 627/627 | 2 | 0.634 (0.632–0.636) | 1 | — | 6 | 0.634 (0.632–0.636) |
| No.2 | 562/562 | 4 | 0.695 (0.692–0.698) | 5 | 0.724[a] (0.721–0.727) | 6 | 0.733[a] (0.730–0.736) |
| No.3 | 367/367 | 8 | 0.747 (0.722–0.752) | 4 | 0.729[b] (0.725–0.733) | 6 | 0.737[b] (0.732–0.742) |
| No.4 | 271/277 | 10 | 0.774 (0.766–0.782) | 4 | 0.751b (0.743–0.759) | 6 | 0.737[b] (0.729–0.745) |
| No.5 | 210/222 | 11 | 0.764 (0.755–0.773) | 4 | 0.760 (0.749–0.771) | 6 | 0.728[b] (0.712–0.739) |
| No.6 | 171/181 | 11 | 0.756 (0.743–0.769) | 5 | 0.785[a] (0.755–0.775) | 6 | 0.746 (0.732–0.760) |
| No.7 | 125/148 | 8 | 0.820 (0.803–0.837) | 4 | 0.817 (0.801–0.833) | 6 | 0.824 (0.808–0.840) |
| **Multicenter testing cohort** | | | | | | | |
| No.1 | 414/414 | 2 | 0.631 (0.628–0.634) | 1 | — | 3 | 0.602[b] (0.599–0.605) |
| No.2 | 359/359 | 4 | 0.689 (0.685–0.693) | 4 | 0.698[a] (0.694–0.702) | 6 | 0.715[a] (0.711–0.719) |
| No.3 | 233/234 | 7 | 0.725 (0.718–0.732) | 4 | 0.715 (0.709–0.721) | 6 | 0.720 (0.714–0.726) |
| No.4 | 181/189 | 8 | 0.790 (0.781–0.799) | 4 | 0.752[b] (0.742–0.762) | 6 | 0.759[b] (0.749–0.769) |
| No.5 | 131/149 | 7 | 0.778 (0.765–0.791) | 4 | 0.757[b] (0.745–0.769) | 6 | 0.754[b] (0.743–0.765) |
| No.6 | 113/128 | 7 | 0.769 (0.750–0.788) | 4 | 0.714[b] (0.695–0.733) | 6 | 0.734 (0.716–0.752) |

[a]The *c*-index of the interested system was higher than the survival path system, with *P* < 0.006
[b]The *c*-index of the interested system was lower than the survival path system, with *P* < 0.006

**Table 4 Hazard ratio and significance of upper node versus lower node at each path bifurcation in the derivation, internal testing, and multicenter testing cohorts**

| Bifurcation node | Derivation cohort | | Internal testing cohort | | Multicenter testing cohort | |
|---|---|---|---|---|---|---|
| | HR (95% CI) | *P* value | HR (95% CI) | *P* value | HR (95% CI) | *P* value |
| $S_{(all;\ ts=1)}$ | 2.33 (1.97–2.75) | <0.001 | 2.58 (2.00–3.32) | <0.001 | 2.34 (1.80–3.05) | <0.001 |
| $S_{(p=1,\ ts=1)}$ | 3.70 (2.78–4.93) | <0.001 | 3.69 (2.40–5.69) | <0.001 | 2.79 (0.171–4.53) | <0.001 |
| $S_{(p=1,\ ts=2)}$ | 3.50 (2.26–5.43) | <0.001 | 2.52 (1.29–4.91) | 0.007 | 5.95 (2.98–11.91) | <0.001 |
| $S_{(p=1,\ ts=3)}$ | 5.25 (2.97–9.31) | <0.001 | 4.02 (1.57–10.31) | 0.004 | 12.27 (4.53–33.26) | <0.001 |
| $S_{(p=1,\ ts=4)}$ | 5.08 (2.24–11.53) | <0.001 | 6.35 (1.51–26.75) | 0.012 | — | —[b] |
| $S_{(p=2,\ ts=1)}$ | 2.73 (2.04–3.66) | <0.001 | 3.31 (2.06–5.32) | <0.001 | 2.47 (1.64–3.73) | <0.001 |
| $S_{(p=2,\ ts=2)}$ | 6.45 (3.29–12.62) | <0.001 | 7.16 (2.00–25.60) | 0.002 | 6.12 (2.69–13.94) | <0.001 |
| $S_{(p=3,\ ts=3)}$ | 5.61 (2.05–15.34) | <0.001 | — | 0.052[a] | — | —[b] |
| $S_{(p=4,\ ts=2)}$ | 4.26 (2.35–7.72) | <0.001 | 3.83 (1.84–7.98) | <0.001 | 3.48 (1.33–9.11) | 0.011 |
| $S_{(p=5,\ ts=3)}$ | 10.35 (3.17–33.82) | <0.001 | — | —[b] | — | —[b] |
| $S_{(p=5,\ ts=4)}$ | 6.89 (1.47–32.14) | 0.005 | — | —[b] | — | —[b] |
| $S_{(p=8,\ ts=4)}$ | 4.45 (1.67–11.85) | 0.003 | 3.67 (1.03–8.71) | 0.048 | — | —[b] |

p: path, ts: time slice
[a]No deaths were recorded in one node and Kaplan–Meier Method with log rank test was utilized
[b]Sample size in one node of the two comparators was <6

**Table 5 The correlation between surgery/ablation and path transfer in KEY nodes**

| Nodes, *n* | With surgery/ablation | | | Without surgery/ablation | | | *P* value |
|---|---|---|---|---|---|---|---|
| | Go up (*n*, %) | Go down (*n*, %) | Died/NS (*n*, %) | Go up (*n*, %) | Go down (*n*, %) | Died/NS (*n*, %) | |
| $S_{(p=3,\ ts=2)}$ | 20 (83.3) | 2 (8.3) | 2 (8.3) | 37 (46.8) | 2 (2.5) | 40 (50.6) | <0.001 |
| $S_{(p=5,\ ts=3)}$ | 11 (84.6) | 1 (7.7) | 1 (7.7) | 20 (50.0) | 5 (12.5) | 15 (37.5) | 0.072[a] |
| $S_{(p=8,\ ts=4)}$ | 13 (81.3) | 2 (12.5) | 1 (6.3) | 16 (45.7) | 12 (34.3) | 7 (20.0) | 0.070[a] |
| $S_{(p=2,\ ts=1)}$ | 79 (71.2) | 24 (21.6) | 8 (7.2) | 50 (15.2) | 203 (61.5) | 77 (23.3) | <0.001 |
| $S_{(p=4,\ ts=2)}$ | 17 (56.7) | 7 (23.3) | 6 (20.0) | 28 (14.2) | 38 (19.3) | 131 (66.5) | <0.001 |

p: path, ts: time slice, NS: no surveillance
[a]Fisher's exact test

**Table 6 Variables and methods of dichotomization for construction of the survival paths**

| Categories and variables | Methods of dichotomization |
|---|---|
| Laboratory tests | |
| Serum AFP level (IU/ml) | <200 vs. ≥200; <400 vs. ≥400 |
| Child-Pugh class | Class B/C vs. class A; class C vs. class A/B |
| Imaging examination | |
| Diameter of main lesion (mm) | ≤50 vs. >50; ≤70 vs. >70; ≤100 vs. >100 |
| Number and size of hepatic lesions | ≤1 lesion/2–3 lesions, $D \leq 30$ mm vs. >3 lesions/2–3 lesions, $D > 30$ mm; <4 lesions vs. ≥4 lesions |
| Vascular invasion | With vs. without |
| Distant metastasis | With vs. without |
| Vascular invasion/N1/M1 | With vs. without |
| Change of lesions | With viable lesion vs. without viable lesion |
| Performance status | 0–2 vs. >2 |

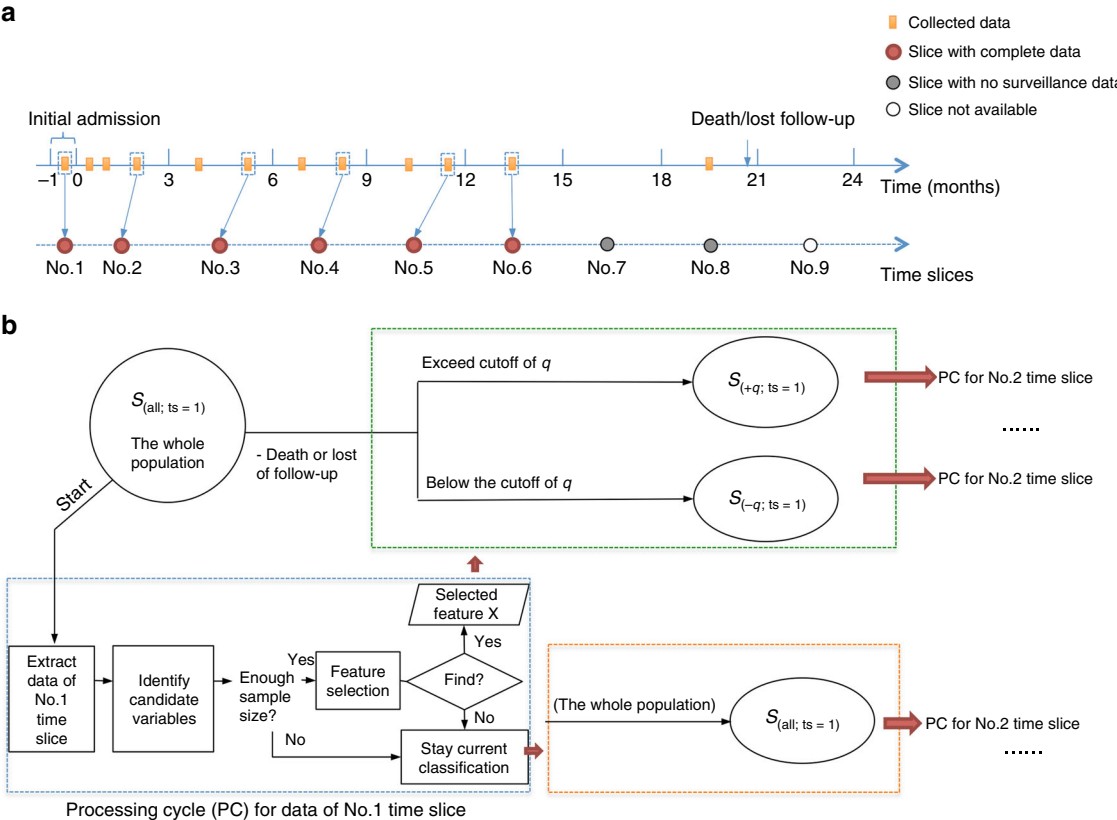

**Fig. 4** Flowchart of study design. The time-series data of HCC patients were converted into data of time slices with constant time interval. Time slices in each case with complete data were enrolled for further analysis (**a**). Data in the first time slice of the whole population would initially undergo the processing cycle (PC) for feature selection and subgroup subdivision. Then data of the next time slice in each subdivided subgroup sequentially undergo PC; the analytical cycles continue until the completion of the last time slice (**b**)

point was set at the time of diagnosis of HCC. An interval of 3 months was utilized and the time ranges (in months) of −1~0, 0.5~3, 3.1~6, 6.1~9, 9.1~12, 12.1~15, 15.1~18, 18.1~21, and 21.1~24 were transformed into 9 consecutive time slices. For variables measured more than once in each time slice, the newest values were selected to be associated with the time slice. The time slice with complete data of the nine variables was defined as slice with complete data. If the data in one time slice were incomplete or unavailable, but the follow-up suggested the patient was still alive, the data in this slice and subsequent slices were regarded as point with no surveillance. If the patient died or lost follow-up, the data in the following time slice were regarded as nonexistent (Fig. 4a).

The primary outcome was OS. For data in the first slice, OS was defined as time from diagnosis to death by any causes. For subsequent time slices, OS was defined as time from image examination of that time slice to death by any causes.

**Feature selection and construction of survival path**. Step 1: We start with using the data at first time slice of the derivation cohort, which is denoted $S_{(\text{all; ts}=1)}$ (Fig. 4b). Univariate analysis with Kaplan–Meier (KM) method was utilized to

identify candidate variables $(X_1, X_2, \ldots, X_p)$. In constructing the survival path, each feature selection process at a specific time slice was considered as an independent experiment. To control the false discovery rate, suppose we have $m$ time slices, the preselected significance level for feature selection in each path was calculated by the formula below[26]:

$$\frac{\alpha}{m} \tag{1}$$

Step 2: Judgment: The sample size required for the Cox proportional hazard regression model with multiple covariates was calculated[27]. If the sample size is larger than the calculated one, the data would proceed to feature selection. Less than calculated sample size will stop any future feature selection process and the group will remain the current classification.

Step 3: Feature selection: All the significant variables detected using the KM method were put into the Cox proportional hazard regression model, which

assumes the hazard as follows,

$$h(t) = h_0(t)\exp\left(\sum_{j=1}^{p} \beta_j x_j\right) \quad (2)$$

where $(x_1, x_2,\ldots, x_p)$ is a vector of $p$ predictor variables, and $\beta_1, \beta_2,\ldots, \beta_p$ are the corresponding regression coefficients, which are the weights given to each variable by the model. The original model included all the candidate variables and was presented as follows:

$$Y = \beta_0 + \beta_1 X_1 + \ldots + \beta_{r-1} X_{r-1} + \varepsilon \quad (3)$$

The backward elimination (BE) procedure was carried out, with the following $p$ tests, $H_{0j} : \beta_j = 0, j = 1, 2, \ldots, p$, the lowest partial $F$-test value $F_l$ corresponding to $H_{0l} : \beta_l = 0$ is compared with the preselected significance values $F_0$. If $F_l < F_0$, then $F_l$ can be deleted and the new original model is:

$$Y = \beta_0 + \beta_1 X_1 + \ldots + \beta_{l-1} X_{l-1} + \beta_{l+1} X_{l+1} + \ldots + \beta_{r-1} X_{r-1} + \varepsilon \quad (4)$$

Then, a stepwise BE procedure was continued, until all $F_l > F_0$, and the model is what we choose. The importance of each variable in the fixed Cox model can be obtained as follows:

$$\Gamma_q = -2\log\left(L_h/L_{h-q}\right) \quad (5)$$

where $L_h$ refers to the likelihood of the fixed model and $L_{h-q}$ refers to the likelihood of model after elimination of the variable $X_q$. The variable eliminated from the model with the maximal change of $-2\log$ likelihood was selected.

Step 4: Based on the selected dichotomized variable $X_q$, the cohort $S_{(all;\ ts=1)}$ can be divided into two subgroups $S_{(-q;\ ts=1)}$ and $S_{(+q;\ ts=1)}$. The data of the two subgroups at the next time slice, i.e., the $S_{(-q;\ ts=2)}$ and $S_{(+q;\ ts=2)}$ will repeat steps 1–3, respectively (Fig. 4b). If there is no variable selected, the cohort stays the current classification and the data in next time slice will repeat step 1–3.

Step 5: Graphic representation: The survival path was constructed and visualized using two-dimensional graph, with the time slices on $x$-axis and median OS time on $y$-axis.

**Statistical analysis**. Pearson $\chi^2$ test was used to compare categorical variables between groups. To compare the efficacy in dynamic prognosis prediction between the survival path method and conventional staging systems, the measurement of $c$-index in each time slice was computed and compared using $Z$ test method. All analyses were performed using SPSS version 20.0 (IBM Corporation, USA) and R version 3.3.2 (The R Foundation for Statistical Computing, 2016).

**Data availability**. All the relevant raw data that support the findings of this study have been deposited in the Dryad Digital Repository (https://datadryad.org//) datasets (doi:10.5061/dryad.pd44k8r). In addition, the authenticity of this data has also been validated by uploading the critical raw data onto the Research Data Deposit public platform (www.researchdata.org.cn), with the approval RDD number as RDDA2018000603.

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

## Acknowledgements

We would like to thank Ms. Juan Nie for providing continuous encouragement to Dr. Lujun Shen in the past 4 years. This work was supported by grants from National High Technology Research and Development Program of China (863 Program) (No. 2012AA022701) and National Natural Science Foundation of China (No. 81703323).

## Author contributions

L.S. and W.L. designed and implemented methodology of survival path. L.S., Q.Z., P.G., J.H., C.L., B.C., Q.C., and T.H. participated in collecting the clinical data for modeling. L.S., Q.Z., J.H, T.P., L.Y., Q.C., and T.H. participated in collecting the clinical data for validation. N.W. and P.W. managed and advised on the project. L.S., Q.Z., P.G., and N.W. wrote the paper.

## Additional information

**Competing interests:** The authors declare no competing interests.

