## [Peer Review File · Nature Communications]

Reviewers' comments:

Reviewer #1 (Remarks to the Author):

This paper explores the use of a “cascading survival map” to capture the dynamics of disease progression and prognosis – with a focus on cancer. The approach involves converting time-series clinical data into a graph that splits at time points based upon the features of the data. A deep forest method is used (that has previously been shown to perform better than other deep learning approaches). This uses a cascading structure of decision trees. The approach extends this by modelling time-series data and is tested on a database of survival data for patients with BCLC stage B hepatocellular carcinoma.

The paper is well written with a clear description of the methodology, making the experiments easy to reimplement. Results are documented in a comprehensive manner with a combination of C-index statistics, Kaplan Maier plots and survival path plots.

The background literature is somewhat lacking and a number of related methods for reasoning about longitudinal data have been omitted. For example, the extensive use of temporal abstraction within the literature (Batal 2009), and dynamic models of disease progression (Tucker 2017). What is more, whilst the method appears reasonable with good motivation, many of the decisions for the approach and choice of parameters need to be better justified (as there are many other approaches that could have been used to explore the issue of modelling time-series with splitting trajectories):

- The method works on regular interval time-series (though it seems the intervals change after remission in the dataset description). As a result, the data is pre-processed by grouping data points into intervals of 3 months. This has resulted in some missing data values as well as some removal of data repeats (relying on the most recent data point). Again, perhaps other methods for dealing with irregular time-series should be discussed and / or explored here.
- Validation is through a simple holdout method – the sample sizes are large but I wonder why not other resampling approaches that test more fully for overfitting and confidence – e.g. cross-validation or bootstrapping.
- Section 3 – step 1: Why must the sample size threshold be set at 20? This needs to be justified (or different variations should be explored).
- Overall the statistics seem sound. My only concern is in Step 2 which makes many repeated tests using Kaplan Meier. Depending how many significance tests are made (I think potentially many), then there is a risk of a high False Discovery Rate. This should be explored – to ensure no adaptation to the test is needed - in a similar way to tests that have been developed within the systems biology community (Reiner 2003).
- In the introduction, the use of random forests are discussed but I cannot see any use of these in the method.
- Results are compared to the Barcelona-Clínic Liver Cancer staging system. This is a reasonable comparison as it is a commonly used baseline but what about other state-of-the-art machine learning approaches? In addition, only one dataset has been explored here. This is a major weakness and I would seriously urge the authors to explore at least simulated data to better understand the parameters / behaviour of their approach on data with different characteristics. It would be very risky to make solid conclusions from one dataset using a simple hold-out (as has been carried out here).

Finally, I believe that the key novelty of the approach is in its ability to explain the underlying sub-paths based upon the features involved in each split. This has not really been explored in any detail and I would suggest some further analysis of how the paths (as in Fig 2) can be used by clinicians to help understand the disease in question.

Batal I, Sacchi L, Bellazzi R, Hauskrecht M. A Temporal Abstraction Framework for Classifying Clinical Temporal Data. AMIA Annual Symposium Proceedings. 2009;2009:29-33.

Reiner, A. Yekutieli, D. and Benjamini, Y. Identifying differentially expressed genes using false

discovery rate controlling procedures, vol. 19 no. 3 2003, pages 368-375 DOI:
10.1093/bioinformatics/btf877

Tucker, A. Yuanxi Li, Y. Garway-Heath, D. Updating Markov models to integrate cross-sectional and longitudinal studies, *Artificial Intelligence in Medicine*, vol 77, March 2017, Pages 23–30

Reviewer #2 (Remarks to the Author):

The authors present a proposal for dynamic prognostication in patients with intermediate stage HCC. The prognostic pathways are derived from a single centre retrospective data set separated into a training and validation set. There are already a number of simple prognostic systems for this patient group apart from BCLC such as the HAP score and the ART score which are not mentioned in the manuscript. There are some major issues which limit the interpretation and application of the proposal.

Introduction

1. It is stated the majority of level 1 evidence focusses on first line treatment. This is not the case and almost all of the first line trials of systemic therapy include patients who may have had surgical or loco-regional therapy

Methods

1. To derive the pathways, nine variables were chosen and the variables were dichotomised. There is no explanation as to how these variables were chosen or how the cut-offs were derived. Some seem have little discriminatory power. For example, 97% of patient had hepatitis B so there seems no reason to make this a factor.

2. Table 1 is not clear; what is meant by the vascular invasion category, with vs without and PVI vs without PVI – are these the same or different? Similarly what is meant by with or without viable lesion – does this mean that no lesion can be seen as per RECIST 1.1 or there is no arterialisation seen as per modified RECIST or some other criteria. Is a greater or less than 10mm enlargement reliable or is it beyond the sensitivity of imaging modality? Why not use conventional RECIST to define progression?

3. It is acknowledged that the patients included had a range of different therapies but these are not defined in the prognostic variables or patient characteristics.

Results

1. Table 2 does not include key characteristics such as performance status, BCLC, metastatic disease, vascular invasion or first treatment given. Performance status in part of the BCLC system and patients have to be performance status 0 to be BCLC B. It is often a challenge to obtain reliable performance status data from retrospective data sets unless it has been systematically recorded. If it has not been collected then the entire paper is undermined since BCLC stage cannot be allocated.

2. The survival path system on figure 2 presents 16 paths all of which, with the exception of path no 1. have very small numbers of patient making intergroup comparison unreliable. The problem is exemplified by the fact that the survival outcomes for these groups varies depending on the time of analysis as shown in figure 3. It is unclear from figure 3 which of the KM curves are significantly different from the others.

Discussion

A major flaw is the clinical applicability of the system. It is highly complex and it is completely unclear how this will inform clinical decision making.

Replies to Reviewer #1:

Q1.This paper explores the use of a “cascading survival map” to capture the dynamics of disease progression and prognosis – with a focus on cancer. The approach involves converting time-series clinical data into a graph that splits at time points based upon the features of the data. A deep forest method is used (that has previously been shown to perform better than other deep learning approaches). This uses a cascading structure of decision trees. The approach extends this by modelling time-series data and is tested on a database of survival data for patients with BCLC stage B hepatocellular carcinoma.

The paper is well written with a clear description of the methodology, making the experiments easy to reimplement. Results are documented in a comprehensive manner with a combination of C-index statistics, Kaplan Maier plots and survival path plots.

Answer: Many thanks for the acknowledgements from the reviewer. The methodology of survival path mapping was based on time-series data conversion using “time slices” and Cox regression based feature selection, which is easy to understand and reimplement. In addition to the comprehensive analysis manner, in this revised manuscript, we also dig more on the clinical implications and applications of this model in patients with BCLC stage B HCC and did further analysis to correlate treatment with specific path bifurcations.

Q2. The background literature is somewhat lacking and a number of related methods for reasoning about longitudinal data have been omitted. For example, the extensive use of temporal abstraction within the literature (Batal 2009), and dynamic models of disease progression (Tucker 2017). What is more, whilst the method appears reasonable with good motivation, many of the decisions for the approach and choice of parameters need to be better justified (as there are many other approaches that could have been used to explore the issue of modelling time-series with splitting trajectories):

References provided by reviewer:

Batal I, Sacchi L, Bellazzi R, Hauskrecht M. A Temporal Abstraction Framework for Classifying Clinical Temporal Data. AMIA Annual Symposium Proceedings. 2009;2009:29-33.

Tucker, A. Yuanxi Li, Y. Garway-Heath, D. Updating Markov models to integrate cross-sectional and longitudinal studies, Artificial Intelligence in Medicine, vol 77, March 2017, Pages 23–30

Answer: Many thanks for this valuable suggestion. We cited the past literatures about the temporal abstraction and Hidden Markov Models in modeling time-series with splitting trajectories in the "Introduction" section of revised manuscript as bellowed:

Several studies have been reported that the Joint Models can describe the change in the prognostic value of a single variable measured over time, while these models do not support multiple dynamic variables, which limit its clinical application. *In the field of machine learning, the methodologies of temporal abstractions and Hidden Markov Models had been utilized for classifying patients with longitudinal data by building splitting trajectories; these models supported multiple variables and achieved better performance compared to modeling just using cross-sectional data. However, these models are considered opaque since internal structure and learned parameters are difficult for interpretation. Moreover, pure pursuit of the precision in prognosis prediction it not that important as an ideal predictive system should also provide clues in treatment planning.*

In terms of the choice of parameters, based on past literature on staging systems and laboratory tests (1-2), as well as the feedback from the reviewer #2 (Q3, Q4, Q6, Reviewer #2), this time, a total of 9 variables were designed, covering imaging results, laboratory tests and performance status; all the variables were dichotomized. Using selected parameters, in the revised manuscript, we re-analyzed our data.

To better justify the choice of parameters, we revised the associated sentences in the "Method" Section as bellowed:

Time-series data on serum tumor markers, biochemical and hematological indices, medical imaging and associated changes (CT and/or MRI) of each patient

were collected. Based on past literatures on staging systems and laboratory tests, a total of 9 variables were designed, covering imaging results, laboratory tests and performance status; all the variables were dichotomized (Table1).

We hope the revisions and justifications could make our model more reasonable.

Table 1. Variables and methods of dichotomization for construction of the survival paths

Categories and variables	Methods of dichotomization
Laboratory Tests	
Serum AFP level (IU/ml)	<200 vs. ≥200; <400 vs. ≥400
Child-Pugh Class	Class B/C vs. Class A; Class C vs. Class A/B
Imaging Examination	
Diameter of main lesion (mm)	≤50 vs. >50; ≤70 vs. >70; ≤100 vs. >100
Number and size of hepatic lesions	≤1 lesion/2-3 lesions, D≤30mm vs. >3 lesions/2-3 lesions, D>30mm; <4 lesions vs. ≥ 4 lesions
Vascular invasion	With vs. Without
Distant Metastasis	With vs. Without
Vascular invasion/N1/M1	With vs. Without
Change of lesions	With viable lesion vs. without viable lesion
Performance Status	0-2 vs. >2

References:

1. European Association For The Study Of The L, European Organisation For R, Treatment Of C. EASL-EORTC clinical practice guidelines: management of hepatocellular carcinoma. J Hepatol 2012;56:908-943.
2. Chen ZH, Hong YF, Lin J, Li X, Wu DH, Wen JY, Chen J, et al. Validation and ranking of seven staging systems of hepatocellular carcinoma. Oncol Lett 2017;14:705-714.

Q3. The method works on regular interval time-series (though it seems the intervals change after remission in the dataset description). As a result, the data is pre-processed by grouping data points into intervals of 3 months. This has resulted in some missing data values as well as some removal of data repeats (relying on the most recent data point). Again, perhaps other methods for dealing with irregular time-series should be discussed and / or explored here.

Answer: Many thanks for this valuable suggestion. Patients with HCC are regularly followed up every one to three months during the treatment, every three months during the first year after complete remission and later every three to six months. Therefore, in this study, we converted the time-series data of patients with BCLC stage B HCC into data of time slices with constant interval of three months. This conversion will result in some missing data.

We have thought about the methods in dealing with irregular time-series, for example, we can extend the interval of time slice when complete remission is achieved, and shorten the interval of time slice for specific patient during intense repeated treatment, e.g. TACE sequentially combined Ablation. The challenge we face is: the treatment strategy adopted in specific node varied greatly (repeated TACE, ablation, and even surgery in different sequence). This requires us to develop a learning algorithm to design personalized time slice for each patient, meanwhile also poses a great challenge on the sample size of study, as shortening the interval of time slice will add bifurcations in the early time period but reduce bifurcations in the late time period.

This study is our initial step in designing survival path model for patients with HCC and our future work will also focus on techniques/methods in dealing with irregular time-series. We have added a paragraph in justifying the reason using three months as an interval in the "Introduction" Section of revised manuscript as bellowed:

Patients with HCC are regularly followed up every one to three months during the treatment, every three months during the first year after complete remission and later every three to six months. Therefore, in this study, we converted the time-series data of patients with BCLC stage B HCC into data of time slices with constant interval of three months, and used a process of "Cox-based feature selection" to select key prognostic features to build the first cascading survival map. This map was further evaluated on its ability for dynamic prognosis prediction and treatment planning.

We also added sentences in the "discussion" section of revised manuscript about it:

The survival paths we constructed for stage B HCC was an initial attempt, and efforts could be made in several aspects to further improve this model/methodology...*The third aspect is to develop methods in dealing with irregular time-series. Converting the time-series data into time slices of three months will result in some missing data, therefore, a learning algorithm that dynamically design the*

interval of time slice in specific node based on the characteristics of data is needed to maximally utilize the data.

Q4.Validation is through a simple holdout method – the sample sizes are large but I wonder why not other resampling approaches that test more fully for overfitting and confidence – e.g. cross-validation or bootstrapping.

Answer: Many thanks for this valuable suggestion. We are also concerned about the risk of overfitting. The cross-validation and bootstrapping are powerful ways for validation.

In this revised manuscript, we enlarged the sample size of internal testing/validation dataset (n=627); we believed external validation is also an effective way to validate our model and therefore in the past year we collected multicentric testing/validation dataset in three independent medical centers in Southern China (n=414) to validate the survival path model.

In the revised manuscript, we present the data on the stability and significance of each bifurcation in derivation cohort, internal testing cohort and multicenter test cohort. The P values of all tests were equal or less than 0.05, indicating the methodology we applied for node bifurcation was reliable (Table 5, revised manuscript).

Table 5. Hazard Ratio and Significance of Upper node versus Lower node at each Path Bifurcation in the derivation, internal testing and multicenter testing cohorts

Bifurcation node	Derivation cohort (n=979)		Internal Testing cohort (n=627)		Multicenter Testing cohort (n=414)	
	HR (95%CI)	P value	HR (95%CI)	P value	HR (95%CI)	P value
S _(all, ts=0)	2.33 (1.97-2.75)	<0.001	2.58 (2.00-3.32)	<0.001	2.34 (1.80-3.05)	<0.001
S _(p=1, ts=1)	3.70 (2.78-4.93)	<0.001	3.69 (2.40-5.69)	<0.001	2.79 (.171-4.53)	<0.001
S _(p=1, ts=2)	3.50 (2.26-5.43)	<0.001	2.52 (1.29-4.91)	0.007	5.95 (2.98-11.91)	<0.001
S _(p=1, ts=3)	5.25 (2.97-9.31)	<0.001	4.02 (1.57-10.31)	0.004	12.27 (4.53-33.26)	<0.001
S _(p=1, ts=4)	5.08 (2.24-11.53)	<0.001	6.35 (1.51-26.75)	0.012	—	— ^b
S _(p=2, ts=1)	2.73 (2.04-3.66)	<0.001	3.31 (2.06-5.32)	<0.001	2.47 (1.64-3.73)	<0.001
S _(p=2, ts=2)	6.45 (3.29-12.62)	<0.001	7.16 (2.00-25.60)	0.002	6.12 (2.69-13.94)	<0.001
S _(p=3, ts=3)	5.61 (2.05-15.34)	<0.001	—	0.052 ^a	—	— ^b
S _(p=4, ts=2)	4.26 (2.35-7.72)	<0.001	3.83 (1.84-7.98)	<0.001	3.48 (1.33-9.11)	0.011
S _(p=5, ts=3)	10.35 (3.17-33.82)	<0.001	—	— ^b	—	— ^b
S _(p=5, ts=4)	6.89 (1.47-32.14)	0.005	—	— ^b	—	— ^b
S _(p=8, ts=4)	4.45 (1.67-11.85)	0.003	3.67 (1.03-8.71)	0.048	—	— ^b

Abbreviations: p, path; ts, time slice

^aNo deaths were recorded in one node and Kaplan-Meier Method with Log rank test was utilized. ^bSample size in one node of the two comparators was less than 6.

In the revised manuscript, we also compared the c-index of survival path system with BCLC staging system, AJCC staging System in the internal testing and multicentric testing cohorts (Table 4, revised manuscript). Our results demonstrated that the survival path model had superior or equal value than BCLC staging system, AJCC staging system and ART score in dynamic prognosis prediction from 3-12 months after initial diagnosis; the results were consistent in testing cohorts and derivation cohort.

Table 4. Comparison of C-index between the Survival Path System, BCLC staging system, AJCC staging system in the internal testing cohort and multicenter testing cohort

Time Slice	Number (Modeling/All)	Survival Path System		BCLC staging System		AJCC staging System	
		Number of nodes	C-index (95% CI)	Number of Classes	C-index (95% CI)	Number of Classes	C-index (95% CI)
Internal Testing Cohort							
No.1	627/627	2	0.634 (0.632-0.636)	1	-	6	0.634 (0.632-0.636)
No.2	562/562	4	0.695 (0.692-0.698)	5	0.724 Φ (0.721-0.727)	6	0.733 Φ (0.730-0.736)
No.3	367/367	8	0.747 (0.722-0.752)	4	0.729* (0.725-0.733)	6	0.737* (0.732-0.742)
No.4	271/277	10	0.774 (0.766-0.782)	4	0.751* (0.743-0.759)	6	0.737* (0.729-0.745)
No.5	210/222	11	0.764 (0.755-0.773)	4	0.760 (0.749-0.771)	6	0.728* (0.712-0.739)
No.6	171/181	11	0.756 (0.743-0.769)	5	0.785 Φ (0.755-0.775)	6	0.746 (0.732-0.760)
No.7	125/148	8	0.820 (0.803-0.837)	4	0.817 (0.801-0.833)	6	0.824 (0.808-0.840)
Multicenter Testing Cohort							
No.1	414/414	2	0.631 (0.628-0.634)	1	-	3	0.602* (0.599-0.605)
No.2	359/359	4	0.689 (0.685-0.693)	4	0.698 Φ (0.694-0.702)	6	0.715 Φ (0.711-0.719)
No.3	233/234	7	0.725 (0.718-0.732)	4	0.715 (0.709-0.721)	6	0.720 (0.714-0.726)
No.4	181/189	8	0.790 (0.781-0.799)	4	0.752 * (0.742-0.762)	6	0.759* (0.749-0.769)
No.5	131/149	7	0.778 (0.765-0.791)	4	0.757* (0.745-0.769)	6	0.754* (0.743-0.765)
No.6	113/128	7	0.769 (0.750-0.788)	4	0.714* (0.695-0.733)	6	0.734 (0.716-0.752)

Footnote: *The c-index of the interested system was lower than the survival path system, with P<0.006; Φ The c-index of the interested system was higher than the survival path system, with P<0.006.

Q5. Section 3 – step 1: Why must the sample size threshold be set at 20? This needs to be justified (or different variations should be explored).

Answer: Many thanks for this question. There is an empirical formula to calculate the sample size in Survival Analysis: sample size= 20× (n of variables). Therefore, suppose if we have only one candidate variable, the sample size threshold is 20. We understand the method is not rigorous enough and in the revised manuscript the sample size required for the Cox proportional hazard model with multiple covariates was calculated through the formula below (1):

$$n = \lambda_{(df, \alpha, 1-\phi)} / \left\{ (1-q) \left(h \sum_{i=1}^p a_i^2 + g \sum_{i=1}^p \left(a_i \sum_{j=1, j \neq i}^p a_j \right) \right) \right\}$$

$$a_i = \sigma_i^2 \beta_i + \rho \sum_{j \neq i} \beta_j$$

$$h = (\sigma^2 + (p-2)\rho) / [(\sigma^2 + (p-1)\rho)(\sigma^2 - \rho)]$$

$$g = -\rho / [(\sigma^2 + (p-1)\rho)(\sigma^2 - \rho)]$$

where *df* refers to degree of freedom and equals to *p*, *φ* refers to the probability of type II error, *q* refers to censored rate, *ρ* refers to the mean of covariance, *β_i* is the corresponding regression coefficient, *σ* refers to the corresponding standard deviation. In estimation of sample size, we assume the covariance of each variable is same, *σ*=1, the power (1-*β*) was set at 0.75, *α* was given above, *β_i* for each variable was set at 0.3, *q* was set at 0.25. If the sample size is larger than the calculated one, the data would proceed to feature selection. Less than calculated sample size will stop any future feature selection process and the group will remain the current classification.

Based on the new formula of sample size calculation, in the revised manuscript, we re-analyzed our data and establish the survival path model.

Reference:

Li RP. Sample size calculation for Cox proportional hazards model. Guangzhou: Sun Yat-sen University; 2009.

Q6.Overall the statistics seem sound. My only concern is in Step 2 which makes many repeated tests using Kaplan Meier. Depending how many significance tests are made (I think potentially many), then there is a risk of a high False Discovery Rate. This should be explored – to ensure no adaptation to the test is needed - in a similar way to tests that have been developed within the systems biology community (Reiner 2003).

Reference provided by reviewer:

Reiner, A. Yekutieli, D. and Benjamini, Y. Identifying differentially expressed genes using false discovery rate controlling procedures, vol. 19 no. 3 2003, pages 368-375 DOI: 10.1093/bioinformatics/btf877

Answer: Many thanks for this kind suggestion. High False Discovery Rate is also our concern. Several measures had been made to control high False Discovery Rate: **1)**

suppose we have m_0 time slices, preselected significance level was set at $\frac{\alpha}{m_0}$. In our

study, the preselected significance level is 0.006; 2) Limited the number of candidate parameters. In our study, nine categories of variables were selected, with limited cutoff values; 3) Only use one parameter (the most impactful one) for node bifurcation at each node. The results in internal and external testing cohorts suggest our results at each bifurcation are quite stable (Table 5; Q4, Reviewer #1)

Q7. In the introduction, the use of random forests are discussed but I cannot see any use of these in the method.

Answer: We are sorry for this misunderstanding as we didn't use random forests/deep forest in our study. We cited the deep forests in our previous manuscript in order to introduce the design of the survival path model. The deep forest model we cited was not used for longitudinal data in the cited study (1). To avoid the potential misunderstanding, in the revised manuscript, we deleted the associated sentence in the "Introduction" section.

Reference:

1. Zhou ZH, Feng J. Deep Forest: Towards An Alternative to Deep Neural Networks. arXiv: 1702.08835v1; 2017.

Q8. Results are compared to the Barcelona-Clinic Liver Cancer staging system. This is a reasonable comparison as it is a commonly used baseline but what about other state-of-the-art machine learning approaches? In addition, only one dataset has been explored here. This is a major weakness and I would seriously urge the authors to explore at least simulated data to better understand the parameters / behaviour of their approach on data with different characteristics. It would be very risky to make solid conclusions from one dataset using a simple hold-out (as has been carried out here).

Answer: Many thanks for this insightful suggestion to compare the survival path model with other state-of-the-art machine learning approaches. Currently, there are no mature machine learning approaches utilized in clinical practice to facilitate dynamic prognosis prediction. On the other hand, we aim to develop an easy-to-understand and simple-to-use tool (the survival path model) for clinicians in the management of HCC; struggling towards this goal will sacrifice the precision of prognosis prediction of the survival path system. Therefore, we believe the state-of-the-art machine learning approaches, which could maximally use the available information, are likely to have better performance than the survival path model. From our perspective, the advantages of the survival path model are data visualization and its simplicity for understanding. Moreover, it's also meaningful to compare the survival path model with current staging and risk scoring systems like BCLC staging system, AJCC staging system and ART score. Therefore, in the revised manuscript, we chose to compare the survival path model with widely utilized prognosis prediction models including BCLC staging system, AJCC staging system and ART score.

We are very grateful for the reviewer's suggestion on using simulated data to better understand the parameters we chose and methodology we use. We find the simulation of parameters is quite complicated as each parameter is closely linked with other parameters in HCC, while the inter-link between each two parameter are undetermined and even could vary at different time slices; on the other hand, the links are very important for the success of building a survival path model. Meanwhile, we found the disease (HCC) we chose is especially suitable for survival path model as the information provided during repeated treatment and surveillance is of great importance for clinicians based on our experience. Therefore, in the revised manuscript, we focused specifically on utilizing survival path model for intermediate stage HCC and carefully designed the parameters in building the survival path model. The model was tested in internal and multi-centric external testing cohorts with good performance (Q4, Reviewer #1). We rewrite the conclusion of our study to make it more rigorous as belowed:

This novel methodology/model could be utilized to facilitate dynamic prognosis prediction and treatment planning for patients with BCLC stage B HCC in the future.

We hope the methodology we used and the conclusion we made in revised manuscript are more rigorous now.

Q9.Finally, I believe that the key novelty of the approach is in its ability to explain the underlying sub-paths based upon the features involved in each split. This has not really been explored in any detail and I would suggest some further analysis of how the paths (as in Fig 2) can be used by clinicians to help understand the disease in question.

Answer: Many thanks for this valuable suggestion. In the revised manuscript, we also made efforts in analyzing the sub-paths of survival path map of the intermediate stage HCC patients, including exploring the features of paths with long-term survival, investigation the relationship between treatment and path transfer, and define incurable path/node.

The detailed analysis is included in the “Results” Section in the revised manuscript and also showed bellowed:

Long-term survival based on the survival paths system

Of the 13 paths constructed, 3 paths leads to long-term survival (>60 months), including No.1, No.2 and No.8. The No.1 and No.2 paths reached long-terms survival when “no viable lesions” was achieved. However, disease progression could occur at any time slice in a small proportion of patients even they are on the paths of “long-term survival”. For the No.8 path, due to the limitation of our sample size, the key factors related to long-term survival fail to be identified.

Treatment and the path transfer

Of all the nodes in the survival path system, 5 nodes went down from bifurcated nodes in the previous time slice and bifurcated in the following time slice. These nodes had unfavorable prognosis and the survival path system might provide guidance. Surgery and ablative therapies are considered aggressive management and therefore we described the proportion of patients receiving surgery/ablation in these nodes (Table 6, revised manuscript). The surgery/ablation rates in $S_{(p=3, ts=2)}$, $S_{(p=5, ts=3)}$, $S_{(p=8, ts=4)}$, $S_{(p=2, ts=1)}$ and $S_{(p=4, ts=2)}$ were 23.3%, 24.5%, 31.4%, 25.2%, 13.2%, respectively; candidates who received surgery/ablation had rates of 83.3%, 84.6%, 81.3%, 71.2% and 56.7% going to the upper node in the next time slice, respectively. We define a node meets both following conditions: 1) median OS time of its upper bifurcated node had 10 months higher than that of the lower bifurcated node, 2) more than 80% patients receiving surgery/ablation went to the upper bifurcated node, as a “chance node”; then the $S_{(p=3, ts=2)}$, $S_{(p=5, ts=3)}$ and $S_{(p=8, ts=4)}$ are candidate nodes.

Table 6. The correlation between surgery/ablation and path transfer in KEY nodes

Nodes, n	With Surgery/Ablation			Without Surgery/Ablation			P Value
	Go up (n,%)	Go down (n,%)	Died/NS (n,%)	Go up (n,%)	Go down (n,%)	Died/NS (n,%)	
$S_{(p=3, ts=2)}$	20 (83.3)	2 (8.3)	2 (8.3)	37 (46.8)	2 (2.5)	40 (50.6)	<0.001
$S_{(p=5, ts=3)}$	11 (84.6)	1 (7.7)	1 (7.7)	20 (50.0)	5 (12.5)	15 (37.5)	0.072*
$S_{(p=8, ts=4)}$	13 (81.3)	2 (12.5)	1 (6.3)	16 (45.7)	12 (34.3)	7 (20.0)	0.070*

$S_{(p=2, ts=1)}$	79 (71.2)	24 (21.6)	8 (7.2)	50 (15.2)	203 (61.5)	77 (23.3)	<0.001
$S_{(p=4, ts=2)}$	17 (56.7)	7 (23.3)	6 (20.0)	28 (14.2)	38 (19.3)	131 (66.5)	<0.001

Abbreviations: p, path; ts, time slice; NS, no surveillance.*Fisher's Exact Test.

Incurable disease based on the survival paths system

Of all the paths in this map, two paths (No.10 and No.13) only have one valid node after bifurcation. We define a node without following valid node after bifurcation and had a median OS time less than 5 months as an "incurable node", then $S_{(p=10, ts=4)}$ is the incurable node.

In the future, we will try to work on establishing a larger database of HCC patients in China; more information on sub-paths will help decode the sequence of parameters to "long-term survival".

Replies to Reviewer #2:

Q1. The authors present a proposal for dynamic prognostication in patients with intermediate stage HCC. The prognostic pathways are derived from a single centre retrospective data set separated into a training and validation set. There are already a number of simple prognostic systems for this patient group apart from BCLC such as the HAP score and the ART score which are not mentioned in the manuscript. There are some major issues which limit the interpretation and application of the proposal.

Answer: Many thanks for providing these important information and suggestions for us. In this revised manuscript, we enlarged the sample size for internal testing dataset (n=627). Besides, we also included a multi-center testing dataset (n=414) from three independent medical centers in southern China to validate the survival path system. The results in the internal and external testing datasets suggest that this model is stable.

The HAP score and ART score are important tools utilized for predicting survival of BCLC stage B HCC patients receiving TACE, and it's our mistake to miss introducing these previous works in our previous manuscript. As comprehensive treatment using transarterial chemoembolization (TACE), ablative therapies, surgery or their combinations are increasingly utilized in the management of intermediate stage HCC (1-2), the survival path model, which was designed based on the clinical large datasets, could have a wider adaptations for patients receiving various strategies of treatments, compared to the HAP and ART score systems.

In this revised manuscript, we included the description of HAP and ART scores in the "Introduction" section:

...The current dynamic prognostication systems including Hepatoma Arterial-embolization Prognostic (HAP) score and Assessment for Retreatment with TACE (ART) score were designed specifically for patients receiving arterial-embolization therapies (7, 8). A novel prognostication system that suitable for HCC patients receiving comprehensive treatments and facilitate dynamic treatment planning is desirable.

We had also included the comparisons between Survival Path System and ART score in dynamic prognosis prediction in our derivation cohort (Figure 3, Table 3, revised manuscript); the results suggested the Survival Path System had superior prognostic value than ART score. The superiority of Survival Path System over ART score could be explained by the fact that most patients in our study received comprehensive treatment rather than embolization therapies alone.

References:

1. Forner A, Gilibert M, Bruix J, Raoul JL. Treatment of intermediate-stage hepatocellular carcinoma. *Nat Rev Clin Oncol* 2014;11:525-535.

2. Lencioni R, Chen XP, Dagher L, Venook AP. Treatment of intermediate/advanced hepatocellular carcinoma in the clinic: how can outcomes be improved? *Oncologist* 2010;15 Suppl 4:42-52.

Introduction

Q2. It is stated the majority of level 1 evidence focusses on first line treatment. This is not the case and almost all of the first line trials of systemic therapy include patients who may have had surgical or loco-regional therapy .

Answer: Many thanks for pointing out our mistake. We have removed the sentence in our revised manuscript.

Methods

Q3. To derive the pathways, nine variables were chosen and the variables were dichotomised. There is no explanation as to how these variables were chosen or how the cut-offs were derived. Some seem have little discriminatory power. For example, 97% of patient had hepatitis B so there seems no reason to make this a factor.

Answer: Many thanks for the suggestions. It's quite challenging to design the variables in the survival path model. As too many variables can increase the False Discovery Rate, while missing key variables will definitely affect the quality of the model. As pointed by the reviewer that some variables seem to have little discriminatory power, while some important variables were missing.

In the revised manuscript, we re-design the variables and re-analyzed the data. Based on past literature on staging systems and laboratory tests (1-2), a total of 9 variables were designed, covering imaging results, laboratory tests and performance status; all the variables were dichotomized (Table 1, revised manuscript).

Table 1. Variables and methods of dichotomization for construction of the survival paths

Categories and variables	Methods of dichotomization
Laboratory Tests	
Serum AFP level (IU/ml)	<200 vs. ≥200; <400 vs. ≥400
Child-Pugh Class	Class B/C vs. Class A; Class C vs. Class A/B
Imaging Examination	
Diameter of main lesion (mm)	≤50 vs. >50; ≤70 vs. >70; ≤100 vs. >100
Number and size of hepatic lesions	≤1 lesion/2-3 lesions, D≤30mm vs. >3 lesions/2-3 lesions, D>30mm; <4 lesions vs. ≥ 4 lesions
Vascular invasion	With vs. Without
Distant Metastasis	With vs. Without
Vascular invasion/N1/M1	With vs. Without
Change of lesions	With viable lesion vs. without viable lesion
Performance Status	0-2 vs. >2

References:

1. European Association For The Study Of The L, European Organisation For R, Treatment Of C. EASL-EORTC clinical practice guidelines: management of hepatocellular carcinoma. J Hepatol 2012;56:908-943.
2. Chen ZH, Hong YF, Lin J, Li X, Wu DH, Wen JY, Chen J, et al. Validation and ranking of seven staging systems of hepatocellular carcinoma. Oncol Lett 2017;14:705-714.

Q4. Table 1 is not clear; what is meant by the vascular invasion category, with vs without and PVI vs without PVI – are these the same or different? Similarly what is meant by with or without viable lesion – does this mean that no lesion can be seen as per RECIST 1.1 or there is no arterialisation seen as per modified RECIST or some other criteria. Is a greater or less than 10mm enlargement reliable or is it beyond the sensitivity of imaging modality? Why not use conventional RECIST to define progression?

Answer: Many thanks for these important questions. In our previous manuscript, the vascular invasion in “with vs. without” includes hepatic vein, venae cava inferior, portal vein invasion and other major vascular vasion; while the vascular invasion in “PVI vs without PVI” specifically refer to portal vein invasion.

We realized that it's not clear in using these Abbreviations without explanations. Moreover, in the conventional staging systems of HCC, portal vein invasion was not separate from other major vascular invasions. Therefore, in the revised manuscript, we discarded the “PVI vs without PVI” and only use the concept of “with vs. without” vascular invasion (Table 1).

Likewise, we realized the potential error and also discarded the variable “greater or less than 10mm enlargement” and use “with or without viable lesion” only in describing the change of lesions.

The reason we use modified RECIST criteria is because ablative therapies (including MWA and RFA) were widely applied in the treatment of HCC in China; the mRECIST criteria has been demonstrated more reliable in monitoring the status of disease in patients receiving ablative therapies than RECIST 1.1 (1). Besides, the mRECIST criteria was also widely utilized in response assessment for TACE (2-3).

References:

- 1: Gordic S, Corcuera-Solano I, Stueck A, Besa C, Argiriadi P, Guniganti P, King M, Kihira S, Babb J, Thung S, Taouli B. Evaluation of HCC response to locoregional therapy: Validation of MRI-based response criteria versus explant pathology. *J Hepatol.* 2017 Dec; 67(6):1213-1221
- 2: Aliberti C, Carandina R, Lonardi S, Dadduzio V, Vitale A, Gringeri E, Zanus G, Cillo U. Transarterial Chemoembolization with Small Drug-Eluting Beads in Patients with Hepatocellular Carcinoma: Experience from a Cohort of 421 Patients at an Italian Center. *J Vasc Interv Radiol.* 2017 Nov;28(11):1495-1502.
- 3: Lei J, Zhong J, Luo Y, Yan L, Zhu J, Wang W, Li B, Wen T, Yang J; Liver Surgery Group. Response to transarterial chemoembolization may serve as selection criteria for hepatocellular carcinoma liver transplantation. *Oncotarget.* 2017 Aug 24;8(53):91328-91342.

Q5. It is acknowledged that the patients included had a range of different therapies but these are not defined in the prognostic variables or patient characteristics.

Answer: Many thank for this kind remind. From our part, an idealized model of survival paths should also include the information of treatment to further divide the nodes at each time slice, which can enhance the precision of prognostication and provide treatment guidance for patients located at each specific node. However, as high-level evidences of treating recurrent or residual tumors of HCC are very limited, in clinical practice, the treatment adopted by patients at specific node varied greatly. Adding these information will significantly increase the number of nodes at each time slice and requires a much larger database than we currently have, which is the reason we did not include it in this study.

We realized that the description of treatment for patients in our previous manuscript is too simple and not clear. Therefore, in the revised manuscript, we added the associated patient characteristics in the “Methods” Section as bellowed:

Most patients (1852/2020, 91.7%) in the above mentioned medical centers received TACE as their first-line treatment; 135 (6.7%) patients received surgical resection as initial treatment and 33 (1.6%) patients refused to received treatment. The subsequent therapies after TACE, which constituted of ablative therapies, surgical resection, targeted therapies or palliative chemotherapy, were adopted based on the decision of the multidisciplinary teams including hepatologists, radiologists and interventional radiologists...

Results

Q6. Table 2 does not include key characteristics such as performance status, BCLC, metastatic disease, vascular invasion or first treatment given. Performance status in part of the BCLC system and patients have to be performance status 0 to be BCLC B. It is often a challenge to obtain reliable performance status data from retrospective data sets unless it has been systematically recorded. If it has not been collected then the entire paper is undermined since BCLC stage cannot be allocated.

Answer: Many thanks for this kind remind. The ECOG PS scores are routinely documented for all inpatient HCC patients in the medical centers included in this study; for most outpatient HCC patients, the ECOG scores are documented in outpatient medical records, which can be looked up through the electronic documentation system. For ECOG score measured more than once in each time slice, the newest values were selected to be associated with the time slice. If the patient lacks the information of ECOG score during the follow-up time slice, the data in the time slice will be regarded as “point with no surveillance”.

All the BCLC stage B patients initially included in our study were scored 0 using ECOG PS score.

Q7. The survival path system on figure 2 presents 16 paths all of which, with the exception of path no 1. have very small numbers of patient making intergroup comparison unreliable. The problem is exemplified by the fact that the survival outcomes for these groups varies depending on the time of analysis as shown in figure 3. It is unclear from figure 3 which of the KM curves are significantly different from the others.

Answer:

Many thanks for informing us the concerns about the comparisons between intergroups. The primary aim of establishing this survival path model is to realize dynamic prognosis tracing of HCC patients, meanwhile expecting it could facilitate treatment planning based on the information of the specific path and further bifurcations. Therefore, the model was not built to make the comparisons between every two subgroups (at certain time slice) significant; the model was built based on the significance and stability of each bifurcation. In the revised manuscript, we present the data on the stability and significance of each bifurcation in derivation cohort, internal testing cohort and multicenter test cohort. The P values of all tests were equal or less than 0.05, indicating the methodology we applied for node bifurcation was reliable (Table 5, revised manuscript).

Table 5. Hazard Ratio and Significance of Upper node versus Lower node at each Path Bifurcation in the derivation, internal testing and multicenter testing cohorts

Bifurcation node	Derivation cohort		Internal Testing cohort		Multicenter Testing cohort	
	HR (95%CI)	P value	HR (95%CI)	P value	HR (95%CI)	P value
S _(all, ts=0)	2.33 (1.97-2.75)	<0.001	2.58 (2.00-3.32)	<0.001	2.34 (1.80-3.05)	<0.001
S _(p=1, ts=1)	3.70 (2.78-4.93)	<0.001	3.69 (2.40-5.69)	<0.001	2.79 (.171-4.53)	<0.001
S _(p=1, ts=2)	3.50 (2.26-5.43)	<0.001	2.52 (1.29-4.91)	0.007	5.95 (2.98-11.91)	<0.001
S _(p=1, ts=3)	5.25 (2.97-9.31)	<0.001	4.02 (1.57-10.31)	0.004	12.27 (4.53-33.26)	<0.001
S _(p=1, ts=4)	5.08 (2.24-11.53)	<0.001	6.35 (1.51-26.75)	0.012	—	^b
S _(p=2, ts=1)	2.73 (2.04-3.66)	<0.001	3.31 (2.06-5.32)	<0.001	2.47 (1.64-3.73)	<0.001
S _(p=2, ts=2)	6.45 (3.29-12.62)	<0.001	7.16 (2.00-25.60)	0.002	6.12 (2.69-13.94)	<0.001
S _(p=3, ts=3)	5.61 (2.05-15.34)	<0.001	—	0.052 ^a	—	^b
S _(p=4, ts=2)	4.26 (2.35-7.72)	<0.001	3.83 (1.84-7.98)	<0.001	3.48 (1.33-9.11)	0.011
S _(p=5, ts=3)	10.35 (3.17-33.82)	<0.001	—	^b	—	^b
S _(p=5, ts=4)	6.89 (1.47-32.14)	0.005	—	^b	—	^b
S _(p=8, ts=4)	4.45 (1.67-11.85)	0.003	3.67 (1.03-8.71)	0.048	—	^b

Abbreviations: p, path; ts, time slice

^aNo deaths were recorded in one node and Kaplan-Meier Method with Log rank test was utilized. ^bSample size in one node of the two comparators was less than 6.

The phenomenon of varied outcomes of different paths depending on the time is thought to be caused by the fact that the patients died or lost follow-up at specific path in the previous time slice was not included in the analysis of next time slice; the patients with regular follow-up and complete clinical data are more likely to be the patients with good therapeutic efficacy and compliance.

In figure 3, we intend to show the general efficacy of prognosis prediction between different models. The significance of bifurcations at specific node was solid (Table 5), and the model was not built on the purpose that every two subgroups at each time slice differ significantly with each other in OS.

In the Figure 3 of revised manuscript, we also added the AJCC staging system and ART class to compare their prognostic values with survival path model.

Discussion

Q8. A major flaw is the clinical applicability of the system. It is highly complex and it is completely unclear how this will inform clinical decision making.

Answer: Many thanks for pointing out this important issue about clinical applications. All the variables included in building this survival path model are available in clinical practice. We can use the Figure 2 to position each patient on the survival path map and this process won't take too long as only one variable was needed at each time slice; the model is best suitable for patients from time slice No.3 to No.5, and therefore normally 3 to 5 variables are needed.

The positioning of each patient on the map can help clinicians to predict his survival time and the following path bifurcations can provide clues in treatment planning:

- 1) In the management of patients on the paths going up, we need to closely pay attention to the key variables that could transfer the patient to unfavorable paths in the following time slices. For example, for patient in $S_{(path=1, time\ slice=2)}$, it is recommended to control the disease with ≤ 1 viable lesion or two to three lesions with maximal tumor size less than 30mm by the end of time slice No.3; complete remission, AFP<400ng/ml and Child-Pugh score A are required to maintain the patient on the No.1 path;
- 2) In dealing of patients with progressive disease (going down), aggressive treatments like surgery or ablative therapies could be utilized for those in "chance nodes" ($S_{(p=3, ts=2)}$, $S_{(p=5, ts=3)}$ and $S_{(p=8, ts=4)}$) as more than 80% patients received aggressive treatment at this node went to the upper bifurcated nodes at next time slice (Table 6; revised manuscript);
- 3) Palliative treatments were recommended for patients in "incurable nodes" ($S_{(p=10, ts=4)}$).

In the "Results" section of revised manuscript, we added results on the analysis of the correlation between treatments and path transfer. We also define and identify the paths of "long term survival", "chance nodes" that worth aggressive treatment, and "incurable nodes" to facilitate the treatment planning of clinicians (Q9, Reviewer #1).

In the "Discussion" section of revised manuscript, we added a paragraph in explaining the applications of this model in decision making:

The survival path model we established can also guide treatment planning for intermediate stage HCC. In the management of patients on the paths going up, we need to closely pay attention to the key variables that could transfer the patient to unfavorable paths in the following time slices. For example, for patient in $S_{(p=1, ts=2)}$, it is recommended to control the disease with ≤ 1 viable lesion or two to three lesions with maximal tumor size less than 30mm by the end of time slice No.3. Complete remission, AFP<400ng/ml and Child-Pugh score A are required to maintain the patient on the

No.1 path. In dealing of patients with progressive disease, aggressive treatments like surgery or ablative therapies could be utilized for those in “chance nodes”, while palliative treatments were recommended for patients in “incurable nodes”.

Table 6 (revised manuscript). The correlation between surgery/ablation and path transfer in KEY nodes

Nodes, n	With Surgery/Ablation			Without Surgery/Ablation			P Value
	Go up (n,%)	Go down (n,%)	Died/N S (n,%)	Go up (n,%)	Go down (n,%)	Died/NS (n,%)	
S _(p=3, ts=2)	20 (83.3)	2 (8.3)	2 (8.3)	37 (46.8)	2 (2.5)	40 (50.6)	<0.001
S _(p=5, ts=3)	11 (84.6)	1 (7.7)	1 (7.7)	20 (50.0)	5 (12.5)	15 (37.5)	0.072*
S _(p=8, ts=4)	13 (81.3)	2 (12.5)	1 (6.3)	16 (45.7)	12 (34.3)	7 (20.0)	0.070*
S _(p=2, ts=1)	79 (71.2)	24 (21.6)	8 (7.2)	50 (15.2)	203 (61.5)	77 (23.3)	<0.001
S _(p=4, ts=2)	17 (56.7)	7 (23.3)	6 (20.0)	28 (14.2)	38 (19.3)	131 (66.5)	<0.001

Abbreviations: p, path; ts, time slice; NS, no surveillance.*Fisher’s Exact Test.

Please inform us your concerns about the application of this model during clinical practice.

REVIEWERS' COMMENTS:

Reviewer #1 (Remarks to the Author):

I believe that the reviewer has covered all of my original comments sufficiently.

Reviewer #3 (Remarks to the Author):

I believe the authors have addressed the reviewers' critique fully and adequately.